# MemoryVLN: Memory-Augmented Vision-Language Navigation

## Abstract

Vision-Language Navigation (VLN) requires embodied agents to interpret natural language instructions and navigate through previously unseen complex environments. Despite recent progress with large-scale pretraining and multimodal foundation models, current VLN systems remain limited by shallow temporal modeling and insufficient spatial understanding, often processing only short observation sequences and lacking structured memory. Such deficiency severely hinders them from capturing critical information from earlier observations and building detailed representations with structured spatial contexts, unable to make reliable decisions. In this work, we present MemoryVLN, a memory-augmented VLN framework that explicitly models temporal, spatial, and trajectory information to enable long-horizon reasoning and comprehensive spatial understanding. It includes a temporal reasoning memory to adaptively distinguish short-term dynamics and long-term context through selective frame retention, a spatial intelligence memory to build detailed scene graphs with object entities and relationships, and a trajectory memory to visualize historical actions as top-down trajectory maps to encode holistic motion history. With information-intensive memory features, MemoryVLN demonstrates new state-of-the-art performance on R2R-CE and RxR-CE benchmarks by outperforming previous methods by 5%-10%, while simultaneously achieving high computing efficiency by consuming far fewer input tokens. It even largely outperforms previous models with panoramic observations by using RGB observations only. Moreover, experiments on cross-dataset generalization and enhanced spatial reasoning further validate the effectiveness of MemoryVLN in capturing long-term temporal dependencies and 3D scene understanding for embodied navigation.

## 1 Introduction

Vision-Language Navigation (VLN) aims to teach embodied agents to understand natural language instructions and navigate through previously unseen environments. Despite recent progress enabled by large-scale pretraining and multimodal foundation models Zhang et al. (2024a); Cheng et al. (2025); Wei et al. (2025a); Wang et al. (2026); Ding et al. (2025), VLN remains highly challenging due to its inherent ambiguity in language understanding, long-horizon temporal dependencies, and the need for accurate spatial reasoning under partial observability. At each timestep, an agent must integrate historical visual observations, linguistic context, and spatial cues to determine the next action. This poses a significant challenge for constructing unified representations that can span long temporal horizons and maintain consistent spatial awareness to support robust decision-making under the requirement of real-time reaction.

Memory mechanism has recently proven effective in a wide range of embodied intelligence tasks such as task planning Lei et al. (2025b;a), 3D question answering Ginting et al. (2025), and robot manipulation Shi et al. (2025); Liu et al. (2025). These approaches demonstrate that memory can help embodied systems accumulate knowledge and learn from experiences by using memory mechanisms to store intermediate states, summarize interaction histories, or encode episodic experiences for better context-aware reasoning. However, unlike manipulation or task planning that mostly rely on storing agent states or current spatial maps, VLN requires a comprehensive combination of long-horizon temporal understanding and structural spatial

relationship reasoning, while a unified spatial-temporal memory system for VLN is hardly explored now. How to effectively design a structured and efficient memory system in VLN remains unexplored.

For *temporal modeling*, existing VLN models that attempt to incorporate memory often rely on shallow or implicit forms of historical aggregation. Most methods Cheng et al. (2025); Wei et al. (2025a) adopt shallow temporal fusion strategies or fixed-length history buffers that capture only short-term dependencies between consecutive frames. As a result, these agents struggle to recall critical information from earlier observations, leading to inefficient exploration, repeated visits, and poor localization in long-horizon trajectories. For *spatial understanding*, most methods Yang et al. (2026); Wei et al. (2025a) fail to maintain structured spatial memory. They typically lack structured representations that explicitly describe object layouts and inter-object relationships, resulting in poor spatial awareness and error-prone decisions in complex scenes. Consequently, the agent lacks the geometric and semantic consistency needed to reason about spatial layouts and previously explored areas. These limitations become more pronounced in monocular RGB-only settings, where the absence of depth or panoramic information further constrains the agent's ability to reconstruct spatial geometry and long-term contexts.

To handle the above limitations, we present MemoryVLN, a memory-augmented vision-language navigation framework that redefines how embodied agents utilize memory for decision-making. Unlike prior VLN models that only cache most recent fixed frames (e.g., 16) as visual observations, MemoryVLN introduces a structured three-stream memory system that explicitly decouples temporal, spatial, and trajectory knowledge. (1) A Temporal Reasoning Memory (TRM) performs adaptive long-horizon summarization to retain the most distinctive visual events; (2) A Spatial Intelligence Memory (SIM) constructs explicit scene graphs with object-level semantics and spatial relations for structured reasoning; and (3) A Trajectory Memory transforms historical actions into a top-down spatial map, providing global geometric awareness. Together, these modules form a hierarchical cognitive pipeline that mimics the episodic, semantic, and procedural memory of human brains to enable long-term reasoning, spatial grounding, and motion awareness under monocular RGB input, which establishes a new direction for memory-based embodied intelligence.

Comprehensive experiments on R2R-CE and RxR-CE benchmarks show that MemoryVLN achieves new state-of-the-art performance by outperforming previous methods by 5%-10%, and even largely outperforms previous models with panoramic observations by using RGB observations only. Moreover, MemoryVLN also demonstrates strong performance on cross-dataset generalization and spatial reasoning benchmarks, validating its ability to understand temporal correlations and scene layouts in 3D environments. Plentiful visualizations further demonstrate the effectiveness of MemoryVLN to dynamically leverage past observations to improve the quality of the decision-making process.

## 2 Related Work

### 2.1 Vision-Language Navigation

Vision-Language Navigation (VLN) tasks Nguyen et al. (2019); Wang et al. (2022); Wu et al. (2020) challenge embodied agents to interpret natural language instructions and navigate toward target locations in previously unseen environments. Early approaches Hong et al. (2021); Chen et al. (2021); Liu et al. (2023) primarily focused on discrete navigation settings where agents move along predefined connectivity graphs and typically rely on panoramic RGB-D observations. Recent research Hong et al. (2022); Wang et al. (2023b); An et al. (2022); Dai et al. (2024) has advanced toward continuous and more realistic environments, enabling agents to predict low-level motion commands. Moreover, several studies Zhang et al. (2025); Cheng et al. (2025); Wei et al. (2025a) have begun exploring monocular RGB-D configurations, further enhancing the realism and applicability of VLN systems.

Spatial perception ability is critical for the VLN task to understand surrounding environmental contexts to make reliable decisions. However, recent works Yang et al. (2026); Wei et al. (2025a) have disclosed that current VLN approaches are deficient in spatial perception such as modeling environmental layouts and building inter-object relationships, which severely hinder VLN approaches from achieving comprehensive understanding. To mitigate this, we introduce spatial intelligence memory, which augments VLN approaches by building spatial scene graphs with detailed spatial captions and object relationships to provide strong

support for navigation. With the rapid development of Vision-Language Models (VLMs), RT-2 (Zitkovich et al., 2023) has first demonstrated the potential of transferring web-scale knowledge from VLMs to generalizable robotic manipulation. Recent works focus on how to fully adapt the power of VLMs to enable stronger VLN performance. DiscussNav Long et al. (2024b), MCGPT Zhan et al. (2024), and Instruct-Nav Long et al. (2024a) leverage expert collaboration or memory graphs for error correction and historical summarization. NavGPT Zhou et al. (2024b) and NavGPT-2 Zhou et al. (2024a) generate step-wise textual scene descriptions or further introduce visual grounding to enhance the trajectory quality. NaVid Zhang et al. (2024a), NaVILA Cheng et al. (2025) and Uni-NaVid Zhang et al. (2025) expand the training data scale to 550k navigation samples, 3-5M real/simulation navigation data and 3.6M multi-task trajectories to finetune the VLN model along with additional general VQA data. StreamVLN Wei et al. (2025a) collects about 1.0M simulation trajectory data and 480K VQA data to ensure the generalizability of the VLN model. More recent methods try to tackle the challenges in VLN from different perspectives. Wei et al. Wei et al. (2025b) propose a rewriting-driven augmentation paradigm to create unseen observation-instruction pairs from simulation data. AdaNav Ding et al. (2025) learns a difficulty-aware reasoning policy to dynamically trigger reasoning during the navigation process. Dynam3D Wang et al. (2025c) leverages language-aligned hierarchical 3D representations such as posed RGB-D images as visual input to train 3D-VLM in navigation action prediction. Aux-Think Wang et al. (2025a) trains models to internalize structured reasoning patterns via CoT supervision, while predicting actions directly without explicit reasoning at test time to minimize computational costs. MonoDream Wang et al. (2026) trains the model to predict latent features of panoramic RGB and depth observations with only monocular inputs. DecoVLN Xin et al. (2026) introduces a state action pair-level corrective finetuning strategy and a adaptive refinement mechanism to collects high quality state-action pairs. Typically, the VLN task inherently requires the model to store long-horizon detailed scene information with high capacity to perform navigation towards the target place. However, current methods mostly lack the ability to efficiently process long-horizon temporal observations while simultaneously capturing fine-grained spatial semantics under a limited computational budget, both of which are central to reliable navigation in complex, previously unseen environments.

## 2.2 Memory for Embodied Intelligence

Memory mechanism has shown its effectiveness across a wide range of tasks including video understanding Song et al. (2024); He et al. (2024), cross-modal retrieval Wang et al. (2025b) and video generation Hong & Xu (2023); Rahman et al. (2023). For embodied intelligence, memory mechanism also plays a crucial role by enabling agents to accumulate experience, recall past interactions, and make context-aware decisions over time. For example, STMA Lei et al. (2025b) proposes a spatial-temporal memory agent for the embodied task planning task. LA-EQA Ginting et al. (2025) introduces a structured memory system for long-term active embodied question answering. Meta-memory Mao et al. (2025) builds a joint semantic-spatial memory retrieval/integration mechanism for robot spatial reasoning. MemoryVLA Shi et al. (2025) presents a perceptive-cognitive memory bank to store low-level/high-level semantics to assist vision-language action models for robot manipulation. However, most above methods focus on robot manipulation/embodied question answering tasks, which rely on robot states, history decisions and partially spatial-temporal information to improve decision quality. It's necessary to develop an advanced memory framework to tackle the unique characteristics of VLN by modeling long-term temporal dependencies and capturing holistic information.

Despite growing interest in memory for embodied tasks, the application of principled memory design to VLN remains limited. Some VLN methods implicitly adopt memory-like mechanisms: NavGPT Zhou et al. (2024b) and NavGPT-2 Zhou et al. (2024a) accumulate step-wise textual scene descriptions as a sequential log, while DiscussNav Long et al. (2024b) and InstructNav Long et al. (2024a) maintain structured records or memory graphs to enable error correction and historical summarization. However, these approaches store only coarse-grained textual snapshots, discarding rich visual and spatial content from navigation history, and typically depend on panoramic inputs that are unavailable in monocular settings. More broadly, the majority of recent end-to-end VLN models Zhang et al. (2024a); Cheng et al. (2025); Wei et al. (2025a); Ding et al. (2025); Wang et al. (2026) resort to simply concatenating a fixed window of the most recent frames (typically 16) as raw visual input to the LLM, without any structured mechanism for long-horizon temporal reasoning or detailed spatial perception. While naively increasing this window could in principle provide richer history, it scales the LLM token budget linearly and quickly becomes computationally prohibitive. Consequently,

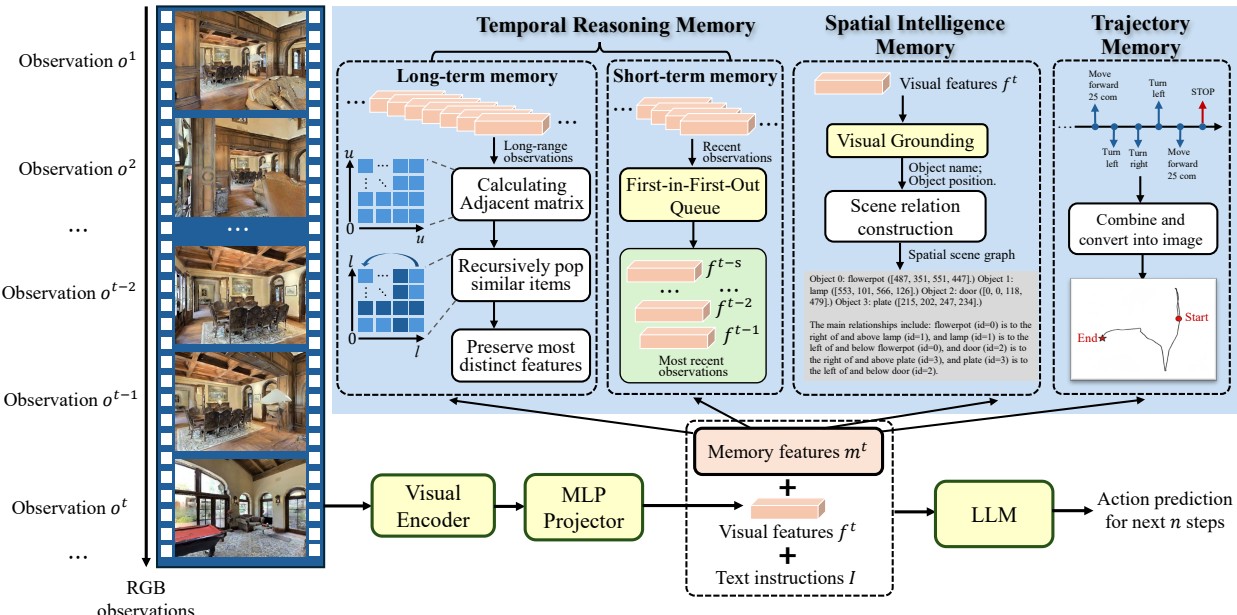

Figure 1: In order to keep useful information of past timesteps while maintaining low inference latency, MemoryVLN decouples the navigation process into vision processing and language processing. Based on past RGB observations, we build temporal reasoning memory, spatial intelligence memory and trajectory memory to model the critical temporal contents, detailed spatial semantics and holistic trajectory information, respectively, to improve the navigation quality. Finally, the concatenated visual feature $x_t$, memory features $m_t$, and the text instructions $I$ are sent into the LLM to predict actions over the next several steps.

such methods remain bound to short visual horizons and lack the structured spatial representations needed for comprehensive scene understanding, all within a practical computing budget. In contrast, MemoryVLN overcomes these limitations through purpose-built memory modules that jointly enable long-horizon temporal reasoning, fine-grained spatial scene understanding, and holistic trajectory modeling, all within a practical computing budget.

# 3 Method

## 3.1 Model Architecture

MemoryVLN supplements the navigation process by constructing memory features with detailed spatial semantics and plentiful temporal contexts to improve the prediction quality. Specifically, MemoryVLN decouples vision processing and language processing into two processes to construct compact memory features. As shown in Fig. 1, a visual encoder and a MLP projector are first used to continuously process the RGB frame $o_t \in R^{3 \times H \times W}$ at each timestep $t$ into compact visual features $x_t \in R^{L \times C}$. Here, $H$ and $W$ denote the input image size (normally $H = W = 384$), $L$ is the number of patches (normally $L = 196$) and $C$ is the channel dimension. Following previous works, we use SigLip Zhai et al. (2023) as the visual encoder, and adopt Qwen2-7B Team et al. (2024) as the LLM decoder. Memory features are then built based on history RGB observations, current detailed spatial context and history predicted actions. To enhance the decision-making ability of MemoryVLN, we introduce a temporal reasoning memory (TRM) to capture distinctive temporal contents within a long duration, a spatial intelligence memory (SIM) which builds detailed spatial scene graphs for the current timestep, and a trajectory memory which models the moving trajectories from a top-down view. Finally, the LLM decoder takes the concatenated visual feature $x_t$, memory features $m_t$, and the text instructions $I$ as inputs to predict the actions $a = \{a_1, \ldots, a_n\}$ for the next $n$ future steps (normally $n = 4$).

### 3.2 Temporal Reasoning Memory

Recent VLN methods Zhang et al. (2024a); Cheng et al. (2025); Wei et al. (2025a); Wang et al. (2026); Ding et al. (2025) mostly employ short temporal fusion or fixed temporal buffers (e.g., 8 or 16 frames) to predict the future actions. However, as consecutive frames are inherently contextually similar, this hand-crafted design easily fails to extract the most important information from abundant visual observations. Besides, it fails to capture distinctive contents across a long-term temporal duration. Naively increasing the length of input frames would drastically increase the computing burden, which may be unavailable in real-world scenarios. To fully model beneficial temporal information of a long temporal range while maintaining efficiency, we introduce temporal reasoning memory (TRM) consisting of short-term memory and long-term memory to model critical contexts of different scales, respectively.

**Short-term memory.** We set a first-in-first-out (FIFO) queue to store the extracted visual features within the most recent $s$ timesteps as short-term memory $m_s = \{x_{t-s}, \ldots, x_{t-1}\}$. It maintains the most relevant visual observations corresponding to the current timestep $t$ to provide closely correlated information.

**Long-term memory.** It aims to preserve the most distinctive features within a long temporal duration to provide sufficient historical information to predict actions. However, as the long-term observations are inherently abundant, it remains a question *how to accurately extract meaningful features from these lengthy observation sequences*. We design long-term memory with a consecutive updating strategy to dynamically model critical information.

Specifically, we first maintain a queue $q_u = \{x_{t-s-u}, \ldots, x_{t-s-1}\}$ with length of $u$ to store the RGB sequence observed by the robot in a long temporal duration. Long-term memory $m_l \in R^{l \times C}$ with $l \ll u$ is then built by selecting representative frames from the queue $q_u$. To keep most distinct features from $q_u$ and discard redundant observations, we design a novel frame selection strategy. Specifically, to identify the content similarity between consecutive frames, we first compute the similarity matrix $\mathcal{U} \in R^{u \times u}$ between adjacent components in the queue $q_u$ in a dot-product manner. Note that $\mathcal{U}$ is a tridiagonal matrix with only non-zero values in the diagonal and sub-diagonal. Given $\mathcal{U}$, we could find the frame pairs that are most similar to each other by recognizing the row and column belonging to the highest value in $\mathcal{U}$, respectively. The left frame within the pair is popped, and the similarity matrix $\mathcal{U}$ is accordingly slimmed by a size of 1 into $R^{(u-1) \times (u-1)}$, by removing the rows and columns belonging to the popped frame. $\mathcal{U}$ is updated by computing the similarities of new neighbors brought by popping a frame. This procedure is repeatedly conducted by removing similar frames until only $l$ frames are kept, which serve as the final elements for long-term memory that contains the most distinctive history observations in $q_u$. Finally, we reduce the spatial size by 2 (e.g., 196 tokens → 49 tokens) for frames in the long-term memory to decrease the computing overhead.

### 3.3 Spatial Intelligence Memory

Spatial perception ability is critical for the VLN task to understand surrounding environmental contexts to make reliable decisions. However, recent works Yang et al. (2026); Wei et al. (2025a) have disclosed that current VLN approaches are deficient in spatial perception such as modeling environmental layouts and building inter-object relationships, which severely hinder VLN approaches from achieving comprehensive understanding. To mitigate this, we introduce spatial intelligence memory, which augments VLN approaches by building spatial scene graphs with detailed spatial captions and object relationships to provide strong support for navigation.

Specifically, we first rely on a visual grounding model to detect all potential objects $O = \{o_1, \ldots, o_w\}$ for the current frame $x_t$. We here use the Grounding DINO model Liu et al. (2024) for grounding, and prompt it with 365 object categories from the Object365 dataset Shao et al. (2019) to cover common objects in everyday scenes. Only objects with detection confidence higher than a predefined threshold are preserved with their class labels and spatial coordinates.

To model the relationships of different objects, we build a scene graph to represent the layout of the current frame. We use a visual backbone (e.g., SigLip) with RoIAlign strategy Ren et al. (2016) to extract spatial features corresponding to each detected object as $F \in R^{w \times C} = \{f_1, \ldots, f_w\}$. The similarities of different objects are computed in a dot-product way based on their feature correspondence as a similarity matrix

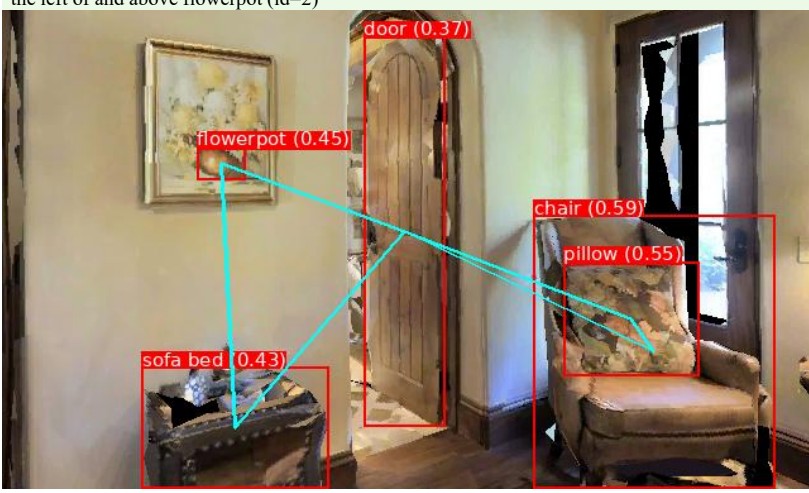

Figure 2: Illustration for the spatial intelligence memory. The top part shows the contexts in spatial intelligence memory in textual format, and the bottom part visualizes the spatial scene graphs with all detected objects and their relationships.

$P \in R^{w \times w}$. A KNN algorithm is then used to select the two most semantically close objects based on $P$ to form two semantic pairs for each object. The scene graph is built by representing objects in each pair as graph nodes, and their relationships as graph edges.

Next, we serialize the scene graph and transform it into textual formats for the LLM. We first give an overview of all objects in the scene as '$This\ frame\ may\ contain\ several\ objects, including$ $\{Obj_1\} : (\{Pos_1\}), \ldots, \{Obj_w\} : (\{Pos_w\})$'. Here, $\{Obj_i\}$ refers to the name of the $i$-th object, and $\{Pos_i\}$ is the spatial location of the $i$-th object. We then present the relationships of each pair in order by referring to a predefined template of '$\{Obj_1\}\ is\ \{Rel\}\ to\ \{Obj_2\}$'. Here, $\{Rel\}$ denotes the spatial relationships of two objects. Finally, we unify the above descriptions, and formulate them as the spatial intelligence memory containing detailed contexts about the entities, locations and relations for objects in current frame. An example of the generated scene graph is given in Fig. 2.

### 3.4 Trajectory Memory

Capturing historical trajectories is critical for robots to model 3D environments and understand historical movements. Previous methods usually rely on panoramic observations to build a panoramic map. However, it's hard to achieve this with RGB observations only. To overcome this limitation, we formulate a trajectory memory based on existing predicted actions to obtain a holistic view of historical trajectories.

Specifically, our goal is to combine historical actions predicted by the VLN model into a top-down trajectory map for the VLN model. A naive solution is to store previous actions and simply sequentially combine them into a text paragraph for the VLN model. However, the outputs may be lengthy for long navigation paths with high abundance and fail to model history trajectories in a panoramic perspective. Instead, we convert historical trajectories into a vision format by setting a blank image as our canvas and depicting the trajectories with directions and distances. Specifically, we begin by labelling the starting point with a solid circle on the blank image, and record its current direction with an invisible arrow. For each predicted action, we change the direction of the arrow accordingly if it belongs to 'turn left/right 15 degrees', and otherwise would draw a line segment following the direction of the arrow. The trajectory memory is completed until the navigation finishes with a 'STOP' action. This formulates an elegant way for robots to understand the

Table 1: Memory efficiency compared with previous works with 16 input frames by default.

| Configurations | Previous works | MemoryVLN | | |
| --- | --- | --- | --- | --- |
| | | TRM | SIM | Traj |
| Temporal range | 16 | $\gg 16$ | 1 | All |
| Information density | Low | **High** | **High** | **High** |
| Spatial granularity | Coarse | Coarse | **Fine** | - |
| Detailed locations | No | No | **Yes** | **Yes** |
| LLM input tokens | $16 \times 196$ | $(8+2) \times 196$ | About 100 | $1 \times 196$ |

trajectories in a top-down view with high information density and flexibility. An illustration of the trajectory memory is given in the bottom right of Fig. 1.

### 3.5 Memory Efficiency

To compare the information coverage and density of previous VLN works with ours, we make a detailed comparison in Tab. 1. One can see that compared with prior VLN models, MemoryVLN delivers substantially higher information density while using far fewer input tokens. Cmpared to recent works like StreamVLN Wei et al. (2025a) and MonoDream Wang et al. (2026), MemoryVLN reduces LLM input tokens by 30% (from $16\times196=3136$ to around $2260\approx1960+100+196$) while simultaneously attending to a much longer temporal range and finer spatial granularity, leading to a more efficient and scalable navigation framework.

## 4 Experiments

### 4.1 Experimental Setup

**Experimental environments.** We evaluate our approach on two public VLN-CE benchmarks: **R2R-CE** Anderson et al. (2018) and **RxR-CE** Ku et al. (2020). The R2R-CE dataset contains 5.6K English trajectories with an average path length of 10 meters, while RxR-CE offers 126K instructions featuring longer and more diverse paths averaging 15 meters. Both datasets adopt a camera horizontal field of view (HFOV) of 79° and require realistic continuous-control indoor navigation. We perform evaluation on the validation-unseen splits of both benchmarks. To evaluate MemoryVLN's spatial scene understanding capabilities, we also conduct experiments on the widely-used **ScanQA** benchmark Azuma et al. (2022) for 3D question answering. It contains more than 41K QA pairs recorded in 800 3D scenes with about 31K unique questions.

**Metrics.** We report standard VLN metrics, including Navigation Error (NE), Success Rate (SR), Oracle Success (OS), and Success weighted by Path Length (SPL), following common practice in prior works.

**Implementation details.** Following recent works, we use LLaVA-Video Zhang et al. (2024b) as the base model with Qwen2-7B Team et al. (2024) as the LLM for fair comparison. We jointly train MemoryVLN on the collection of R2R, RxR and EnvDrop benchmarks for one epoch. During the warm-up phase, we use Adam as the optimizer and apply a peak learning rate of 2e-5 for the language model and 5e-6 for the vision encoder. We set the length of short-term memory as 8 and the length of long-term memory as 8, encompassing a total of 16 frames. MemoryVLN predicts an action chunk with 4 actions for robot execution.

### 4.2 Main Results

**Comparison on VLN-CE R2R and RxR benchmarks.** We compare MemoryVLN with existing VLN-CE models on R2R-CE and RxR-CE benchmarks. These baselines include waypoint predictor-based models and navigation large models. As shown in Tab. 2, by only taking monocular RGB observations as input, our MemoryVLN outperforms all existing models on R2R-CE and RxR-CE benchmarks, even though some previous methods take multiple modalities such as RGB observations, panoramic observations, depth images as inputs. Especially, MemoryVLN outperforms the top waypoint predictor-based model HNR Wang et al. (2024) by 4.8% & 13.8% in Success Rate and 12.0% & 17.3% in Success weighted by Path Length on R2R-CE and RxR-CE benchmarks, respectively. Compared to previous state-of-the-art single RGB-based VLN models such as StreamVLN Wei et al. (2025a) and AdaNav Ding et al. (2025), MemoryVLN achieves

Table 2: Main comparison with prior methods on the Val-Unseen split of R2R-CE and RxR-CE. ∗ indicates methods using an external waypoint predictor. The best and second-best values are labelled with **bold** and underline formats.

| Method | Observation | | | | R2R-CE Val-Unseen | | | | RxR-CE Val-Unseen | | | |
|---|---|---|---|---|---|---|---|---|---|---|---|---|
| | S.RGB | Pano. | Depth | Odo. | NE↓ | OS↑ | SR↑ | SPL↑ | NE↓ | SR↑ | SPL↑ | nDTW↑ |
| *VLN methods with multiple input modalities* | | | | | | | | | | | | |
| GridMM* Wang et al. (2023b) | | ✓ | ✓ | ✓ | 5.11 | 61.0 | 49.0 | 41.0 | - | - | - | - |
| DreamWalker* Wang et al. (2023a) | | ✓ | ✓ | ✓ | 5.53 | 59.0 | 49.0 | 44.0 | - | - | - | - |
| ETPNav* An et al. (2024) | | ✓ | ✓ | ✓ | 4.71 | 65.0 | 57.0 | 49.0 | 5.64 | 54.7 | 44.8 | 63.3 |
| HNR* Wang et al. (2024) | | ✓ | ✓ | ✓ | 4.42 | 67.0 | 61.0 | 51.0 | 5.50 | 56.3 | 46.7 | 63.5 |
| InstructNav Long et al. (2024a) | | ✓ | ✓ | ✓ | 6.89 | - | 31.0 | 24.0 | - | - | - | - |
| LAW Raychaudhuri et al. (2021) | ✓ | | ✓ | ✓ | 6.83 | 44.0 | 35.0 | 31.0 | 10.90 | 8.0 | 8.0 | 38.0 |
| AO-Planner Chen et al. (2025) | | ✓ | ✓ | | 5.55 | 59.0 | 47.0 | 33.0 | 7.06 | 43.3 | 30.5 | 50.1 |
| *VLN methods only keeping most recent RGB frames as visual observations* | | | | | | | | | | | | |
| NaVid Zhang et al. (2024a) | ✓ | | | | 5.47 | 49.0 | 37.0 | 35.0 | - | - | - | |
| Uni-NaVid Zhang et al. (2025) | ✓ | | | | 5.58 | 53.5 | 47.0 | 42.7 | 6.24 | 48.7 | 40.9 | - |
| NaVILA Cheng et al. (2025) | ✓ | | | | 5.22 | 62.5 | 54.0 | 49.0 | 6.77 | 49.3 | 44.0 | 58.8 |
| StreamVLN Wei et al. (2025a) | ✓ | | | | 4.98 | 64.2 | 56.9 | 51.9 | 6.22 | 52.9 | 46.0 | 61.9 |
| AdaNav Ding et al. (2025) | ✓ | | | | 5.01 | 66.6 | 60.2 | 50.0 | 6.21 | 60.5 | 49.8 | 62.2 |
| Dynam3D Wang et al. (2025c) | ✓ | | | | 4.59 | 64.4 | 59.8 | 54.4 | - | - | - | - |
| Aux-Think Wang et al. (2025a) | ✓ | | | | 6.08 | 60.0 | 54.8 | 46.9 | 6.24 | 61.9 | 52.2 | 40.2 |
| MonoDream Wang et al. (2026) | ✓ | | | | 5.45 | 61.5 | 55.8 | 49.1 | 6.38 | 49.4 | 40.9 | - |
| DecoVLN Xin et al. (2026) | ✓ | | | | 5.01 | 63.5 | 56.3 | 50.5 | 5.73 | 54.2 | 46.3 | 63.5 |
| MemoryVLN | ✓ | | | | **4.32** | **68.1** | **65.8** | **63.0** | **4.01** | **70.1** | **64.0** | **75.9** |

Table 3: Cross-dataset performance on the RxR-CE ValUnseen split, without training on RxR-CE training set.

| Method | RxR-CE Val-Unseen | | | |
|---|---|---|---|---|
| | NE↓ | OS↑ | SR↑ | SPL↑ |
| CM2 Georgakis et al. (2022) | 8.98 | 25.3 | 14.4 | 9.2 |
| Seq2Seq Krantz et al. (2020) | 11.8 | 5.02 | 3.51 | 3.43 |
| CMA Krantz et al. (2020) | 11.7 | 10.7 | 4.41 | 2.47 |
| NaVid Zhang et al. (2024a) | 8.57 | 32.2 | 21.3 | 20.0 |
| NaVILA Cheng et al. (2025) | 8.96 | 43.4 | 32.5 | 26.8 |
| Uni-NaVid Zhang et al. (2025) | 8.08 | 40.9 | 29.5 | 28.1 |
| MonoDream Wang et al. (2026) | 8.57 | 35.9 | 25.1 | 21.6 |
| AdaNav Ding et al. (2025) | 8.25 | 48.7 | 38.8 | 31.2 |
| MemoryVLN | **8.08** | **52.3** | **43.5** | **37.2** |

an absolute improvement upon Success Rate of 5.6% & 8.9% on R2R-CE, and 9.6% & 17.2% on RxR-CE, respectively. These results demonstrate the advantage of MemoryVLN by extracting distinct representations from past visual observations as auxiliary signals, while still maintaining comparable computational overhead to other methods.

**Cross-dataset evaluation.** We test the cross-dataset generalization ability of MemoryVLN by training it upon R2R benchmark and evaluating upon the RxR benchmark in a zero-shot manner. As shown in Tab. 3, MemoryVLN surpasses previous VLN models by a large margin, which verifies the effectiveness of MemoryVLN on generalizing across different scenarios.

**Spatial scene understanding.** Robust spatial scene understanding is crucial for robots to perform navigation. We evaluate MemoryVLN on the ScanQA benchmark that is widely used for 3D question answering to test its spatial understanding capacity. As shown in Tab. 4, MemoryVLN notably outperforms previous methods on this challenging benchmark, which verifies its effectiveness in understanding the entities and their relationships in spatial scenes.

Table 4: Evaluation of spatial scene understanding performance on the ScanQA Validation split.

| Method | ScanQA Validation | | | | |
| --- | --- | --- | --- | --- | --- |
| | Bleu-4↑ | Rouge↑ | Cider↑ | Meteor↑ | EM↑ |
| Chat-3Dv2 Huang et al. (2024) | 14.0 | - | 87.6 | - | - |
| Scene-LLM Fu et al. (2024) | 12.0 | 40.0 | 80.0 | 16.6 | 27.2 |
| LEO Huang et al. (2023) | 13.2 | 49.2 | 101.4 | 20.0 | 24.5 |
| Uni-NaVid Zhang et al. (2025) | - | 45.74 | 94.72 | 19.24 | 28.01 |
| NaviLLM | 12.0 | 38.4 | 75.9 | 15.4 | 23.0 |
| NaVILA Cheng et al. (2025) | 15.2 | 48.3 | 99.8 | 19.6 | 27.4 |
| StreamVLN Wei et al. (2025a) | 15.7 | 48.3 | 100.2 | 19.8 | 28.8 |
| AdaNav Ding et al. (2025) (64f) | 16.7 | 50.6 | 102.8 | 21.3 | 29.6 |
| **MemoryVLN** | **18.2** | **53.1** | **107.5** | **25.1** | **33.4** |

Table 5: Ablation on the effectiveness of each component in MemoryVLN.

| Configurations | NE↓ | OS↑ | SR↑ | SPL↑ |
| --- | --- | --- | --- | --- |
| W/o TRM | 4.86 | 63.2 | 60.7 | 56.2 |
| - W/o short-term memory | 4.76 | 64.8 | 62.4 | 59.2 |
| - W/o long-term memory | 4.54 | 66.2 | 63.7 | 61.5 |
| W/o SIM | 4.52 | 66.5 | 64.2 | 61.7 |
| W/o trajectory memory | 4.41 | 67.2 | 64.6 | 62.1 |
| MemoryVLN | **4.32** | **68.1** | **65.8** | **63.0** |

## 4.3 Ablation Study

Unless otherwise stated, we perform ablation experiments on the unseen set of VLN-CE R2R benchmark.

**Effectiveness of each proposed component.** As shown in Tab. 5, deleting any of the TRM, SIM and trajectory memory in MemoryVLN leads to a notable performance drop on VLN-CE R2R, where removing TRM brings the most performance decline. This demonstrates that our proposed memory components could notably supplement the VLN model with critical contexts from complementary views, and temporal dependencies are most critical for the VLN model with information-intensive history observations. Furthermore, we investigate the importance of short-term and long-term memory in TRM. We find that disabling each causes considerable performance drop, and the accuracy loss is more evident by removing short-term memory than deleting long-term memory. This validates that short-term temporal information is more crucial for VLN models to capture closely related contexts, while long-term information is essential for retaining historical scene understanding across the full navigation trajectory.

Table 6: Ablation on the input frames of TRM.   Table 7: Ablation on window sizes $\{w_s, w_l\}$ of short-term memory and long-term memory, with $w_s + w_l = 16$.

| Frame(s) | NE↓ | OS↑ | SR↑ | SPL↑ |
| --- | --- | --- | --- | --- |
| 1 | 4.86 | 63.2 | 60.7 | 56.2 |
| 8 | 4.54 | 66.8 | 64.1 | 61.7 |
| 12 | 4.43 | 67.2 | 64.7 | 62.3 |
| 16 | 4.32 | 68.1 | 65.8 | 63.0 |
| 24 | 4.30 | 68.3 | 66.2 | 63.3 |
| 32 | **4.27** | **68.5** | **66.2** | **63.4** |

| Window size | NE↓ | OS↑ | SR↑ | SPL↑ |
| --- | --- | --- | --- | --- |
| $\{4, 12\}$ | 4.37 | 67.6 | 65.2 | 62.1 |
| $\{6, 10\}$ | 4.34 | 67.8 | 65.0 | 62.3 |
| $\{8, 8\}$ | **4.32** | **68.1** | **65.8** | **63.0** |
| $\{10, 6\}$ | 4.37 | 67.6 | 65.4 | 62.8 |
| $\{12, 4\}$ | 4.39 | 67.8 | 65.2 | 62.4 |

**Ablation for the input frames of TRM.** We test the effects for the length of RGB frames contained within the TRM and summarize the results in Tab. 6. We observe that as input frames grow from 1 to 32, the performance continually exhibits an upward trend, which demonstrates that historical RGB observations

can always offer beneficial information for the VLN model. We further notice that the performance grows fast when input frames are relatively fewer (<16), and gradually slows down when frames further increase. This demonstrates that the benefits brought by additional contexts gradually diminish as frames increase. Considering the compute-performance trade-off and for fair comparison with previous methods, we maintain the same number of input frames (16) for MemoryVLN compared to previous works.

**Configurations for the window size of short-term and long-term memory.** Keeping the total input frames as 16 unchanged, we test the effects for different window sizes of short-term and long-term memory in Tab. 7. We observe that as window size of short-term memory increases from 4, the performance continually rises and reaches a peak when it equals 8. Further increasing the window size of short-term memory hurts the performance, which demonstrates that long-term contexts contain irreplaceable semantics for VLN models. We set the window sizes as {8, 8} by default.

Table 8: Ablation on the frame selection strategy of long-term memory.

| Type | Strategy | NE↓ | OS↑ | SR↑ | SPL↑ |
|---|---|---|---|---|---|
| Merging | Merging similar ones | 4.48 | 64.8 | 63.4 | 61.7 |
| | K-means | 4.58 | 65.4 | 63.7 | 61.2 |
| Selection | Random | 4.46 | 66.1 | 64.0 | 61.7 |
| | Most recent | 4.41 | 67.2 | 64.9 | 62.2 |
| | Motion-based | 4.40 | 67.2 | 64.9 | 62.4 |
| | Ours | **4.32** | **68.1** | **65.8** | **63.0** |

Table 9: Ablation on the information contained in SIM.

| Configurations | NE↓ | OS↑ | SR↑ | SPL↑ |
|---|---|---|---|---|
| W/o all object information | 4.41 | 67.4 | 65.2 | 62.4 |
| - W/o object name | 4.36 | 67.7 | 65.7 | 62.6 |
| - W/o object position | 4.35 | 67.6 | 65.8 | 62.8 |
| W/o scene relations | 4.45 | 67.1 | 64.9 | 62.1 |
| MemoryVLN | **4.32** | **68.1** | **65.8** | **63.0** |

**Analysis for the frame selection mechanism in long-term memory.** Tab. 8 ablates the choices of how to select frames in the long-term memory. These strategies are divided into two categories including frame merging which consecutively merges similar frames within the queue, and frame selection which only preserves distinct frames within the queue. Generally, we find that frame merging strategies perform worse than frame selection strategies. We speculate that frame merging destroys the spatial distributions of RGB frames and makes VLN models difficult to understand. For frame selection strategies, compared to random selection, selecting the most recent frames and motion-based selection (keep frames with high optical flow values), our strategy performs best, which verifies its effectiveness to keep most informative frames under a limited frame budget.

**Analysis for the configurations of SIM.** Tab. 9 analyzes the importance of different contexts in SIM. We notice that either removing object information or eliminating scene relations from SIM leads to a considerable performance drop. We further validate the importance of different components in object contexts including object name and object spatial position. We observe that removing each decreases the performance, where removing object names hurts the performance more. This reflects that MemoryVLN can notably overcome the limitation of previous VLN approaches in spatial understanding by building detailed spatial graphs to model spatial correlations.

**Analysis for the design of trajectory memory.** We compare five trajectory memory variants in Tab. 10. (1) *Textual descriptions* (SR=60.2%) record past actions as a sequential text paragraph fed to the LLM. (2) *3D trajectory structure* (SR=62.4%) reconstructs the 3D path by accumulating depth-estimated displacements at each step. (3) *Topological map* (SR=64.2%) builds a graph of visited waypoints connected by edges. (4) *Pose vectors* (SR=58.1%) directly supply raw coordinate-and-heading tuples as numerical tokens. (5) *Ours* (SR=65.8%) renders the full navigation history onto a blank canvas as a top-down trajectory image. The results show a clear performance hierarchy. Pose vectors perform worst, as bare numerical sequences carry little spatial intuition for the LLM and provide no visual structure. Textual descriptions suffer from verbosity on long paths and fail to encode directional geometry in a spatially grounded form. The 3D structure improves upon these but is sensitive to accumulated depth noise across steps. The topological map captures positional coverage yet discards direction and step distance, limiting orientation-aware reasoning. Our top-down visual trajectory achieves the best result by simultaneously encoding direction, displacement, and the overall motion shape in a single compact image that is naturally interpretable by

Table 10: Ablation on different trajectory memory designs.

| Configurations | Text | 3D structure | Topological map | Pose vector | Ours |
|---|---|---|---|---|---|
| SR(%) | 60.2 | 62.4 | 64.2 | 58.1 | **65.8** |

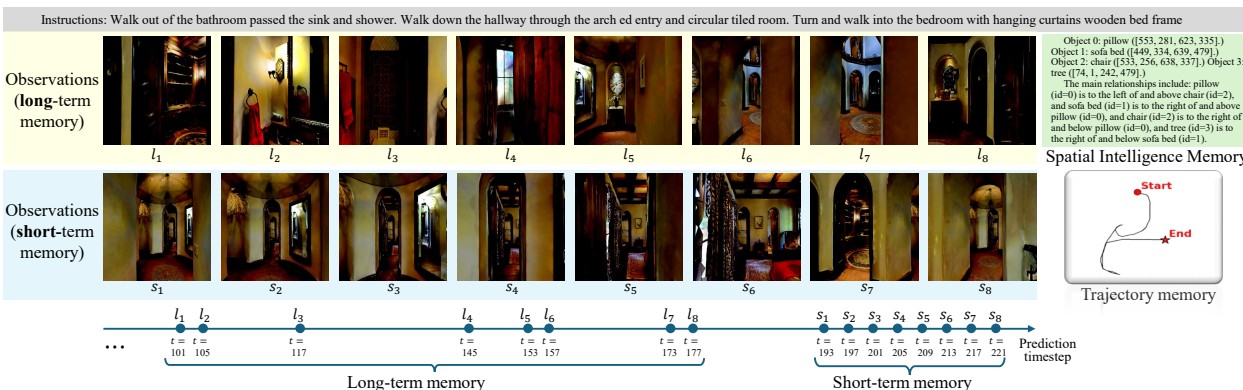

Figure 3: Visualization for the contents in long-term memory, short-term memory, spatial intelligence memory and trajectory memory to verify how MemoryVLN boosts navigation performance.

the visual backbone, confirming that a visual spatial format is the most effective representation for holistic trajectory understanding.

## 4.4 Visualizations

**Analysis of contents in memory.** We visualize the contents in the short-term memory, long-term memory, spatial intelligence memory and trajectory to illustrate how MemoryVLN improves navigation performance. The results are shown in Fig. 3. We observe that frames in the long-term memory distribute unevenly across the timeline, whose contents largely differ from each other. This verifies that long-term memory could preserve the most distinct contexts from long-range observations. The short-term memory stores the most recent observations corresponding to current timestep, which provides strong support for MemoryVLN with the most correlated contexts. The spatial intelligence memory summarizes the key objects and their relations based on their semantic correspondence, which complements the drawback of current VLN approaches in spatial reasoning. The trajectory memory depicts the history trajectory in a top-down view and provides an intuitive way for the model to locate its current position and history trajectories.

**Analysis for the navigation process of MemoryVLN.** We compare the improvement on navigation process of MemoryVLN with our baseline by visualizing their trajectories and RGB observations of intermediate steps in Fig. 4. By equipping memory contexts, MemoryVLN could successfully reach the target position with the shortest path, while the baseline usually consumes more navigation steps and even fails to recognize the target place when it has reached there. More examples can be found in the appendix.

**Analysis of the information density in memory contexts.** Fig. 5 compares the attention distribution of LLM in the VLN model over different input tokens. The inputs are composed of three types of tokens including text instructions, visual tokens and preceding action tokens generated by the LLM. The visual tokens are mostly composed of our proposed memory contents. As revealed by previous studies Chen et al. (2024); Yang et al. (2025), vision-language models mostly pay major attention to input text tokens (>60%) in early layers. Though the visual tokens (thousands of tokens) consume the majority of input tokens compared to text tokens (usually tens of tokens), the attention scores received by visual tokens are rather sparse (<10%). This indicates that the visual tokens are considerably information-redundant and are less informative than text tokens. However, as shown in Fig. 5, MemoryVLN notably improves the attention scores of visual tokens by 3.4× compared to our baseline, which demonstrates that the memory design empowers input visual tokens to contain more distinctive contexts than our baseline by selecting

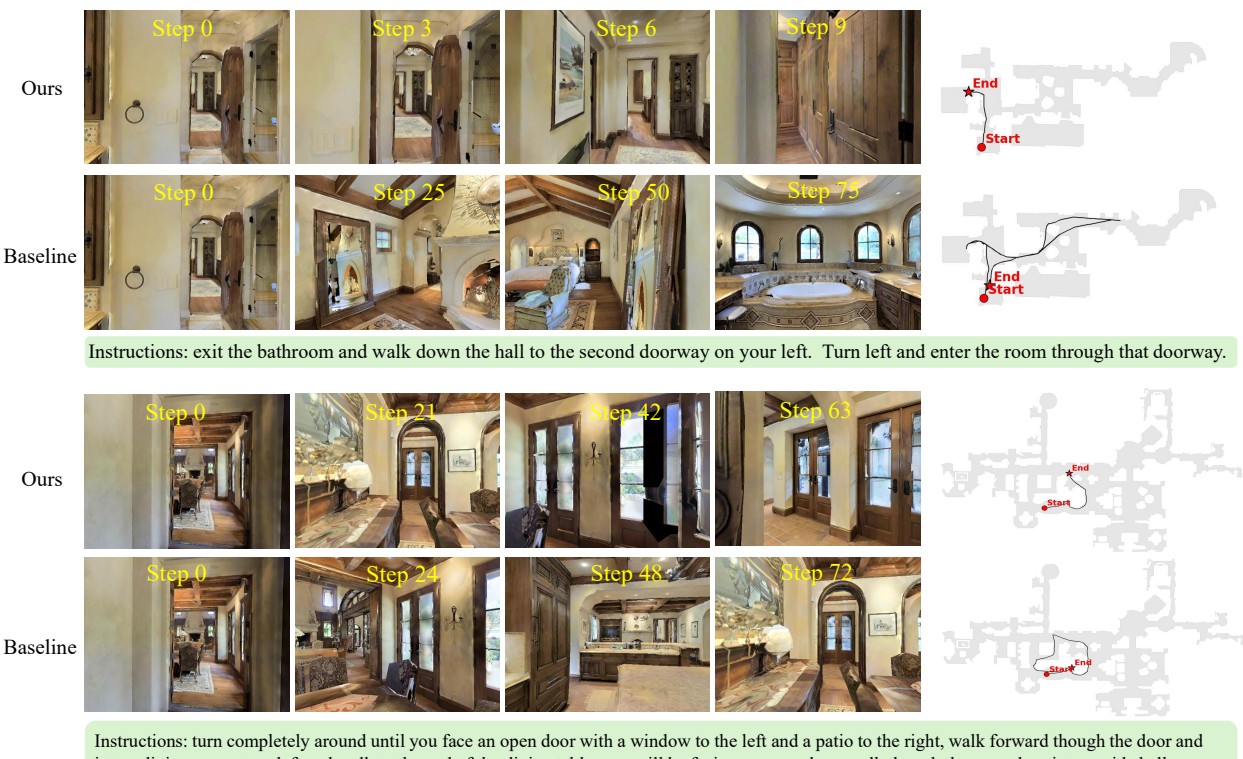

Figure 4: Comparison of the trajectories and intermediate observations of our model against our baseline (no memory).

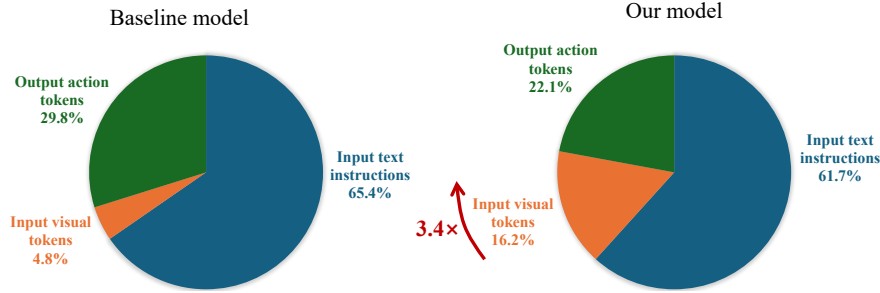

Figure 5: Comparison of attention score distribution of our model against our baseline (no memory).

informative and representative observations from historical contents. This strongly verifies the effectiveness of our memory design.

## 5 Conclusion

In this paper, we presented MemoryVLN, a memory-augmented vision-language navigation framework designed to enhance long-horizon reasoning and spatial understanding for embodied agents. Unlike prior approaches that simply rely on the most recent frames as visual observations, MemoryVLN explicitly models temporal, spatial, and trajectory cues through three complementary memory modules. Comprehensive experiments on R2R-CE and RxR-CE benchmarks show MemoryVLN achieves new state-of-the-art results among RGB-only VLN systems, demonstrating its effectiveness in improving both success rate and path efficiency. Plentiful visualizations validate the effects of memory contexts in preserving distinct information. The ablation analyses confirm that each memory component contributes meaningfully to navigation performance.

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

# A  Appendix

---

**Example for input texts of the LLM**

**User**: You are an autonomous navigation assistant. Your task is to <Instruction>. Devise an action sequence to follow the instruction using the four actions: TURN LEFT by 15 degrees or TURN RIGHT by 15 degrees, MOVE FORWARD by 25 centimeters, or STOP. These are your historical observations: <memory_trm>. In the current scene, <memory_sim>. Here are your historical trajectories <memory_tm>. You can spot <image>.
**Assistant:**  MOVE FORWARD by 25 centimeters, TURN LEFT, TURN RIGHT and MOVE FORWARD by 25 centimeters.
...

*[Here, <Instruction> is the input special token which will be replaced as the encoded tokens of textual instructions for the robot. <Memory_trm> is the special token that will be replaced as the visual tokens for contents in the temporal reasoning memory. <memory_sim> is the special token that will be replaced as the encoded tokens for the textual descriptions of spatial intelligence memory. <memory_tm> is the special token that will be replaced as the encoded visual tokens for the contents in the trajectory memory. <image> is the special token that will be replaced as the encoded visual tokens of the current RGB observation]*

---

## A.1  Chat Template for VLN Models

We illustrate the chat template for our MemoryVLN model as above. The VLN model alternates between perceiving the environment and generating actions conditioned on those observations. The observations and text instructions are interleaved and combined as inputs for the VLN model. Here, <Instruction>, <Memory_trm>, <memory_sim> , <memory_tm> and <image> are the special tokens that will be replaced as the corresponding contexual tokens. In each step, the VLN model predicts the next 4 actions which will be sent for the robot for navigation. Memory-related tokens are only appended after the first step.

Table 11: Ablation on the number of prediction steps.

| Step(s) | NE↓ | OS↑ | SR↑ | SPL↑ |
|---|---|---|---|---|
| 1 | 4.42 | 67.0 | 64.9 | 62.2 |
| 2 | 4.36 | 67.8 | 65.4 | 62.6 |
| 4 | **4.32** | 68.1 | **65.8** | **63.0** |
| 6 | 4.34 | **68.2** | **65.8** | 62.9 |
| 8 | 4.39 | 37.5 | 65.4 | 62.7 |

## A.2  More Ablations

Tab. 11 ablates the performance of MemoryVLN when varying the number of predicted actions per step. We notice that the performance continues to increase as the predicted actions rise from 1, and reaches a peak when the number of predicted actions equals 4. Further increasing the number of predicted actions would hurt the performance. We conclude that modeling RGB observations at each step could provide enough support for a few future predictions. However, the VLN model still requires a relatively dynamic environment observation while each RGB observation could only cover short-term action predictions.

### A.3   More Visualizations

We provide more trajectories and intermediate observations of our model against our baseline in Fig. 6. We notice that compared to our baseline, MemoryVLN always consumes shorter steps while our baseline may take longer decision steps and even reach the maximum limit (75 steps) with navigation failure. Our baseline always wanders around a certain place with no meaningful decisions, and sometimes fail to trigger the 'STOP' signal even when having reached the target place. Instead, our MemoryVLN behaves more intelligently by receiving highly informative contexts to support the decision-making process, which verifies the effectiveness of our memory system.

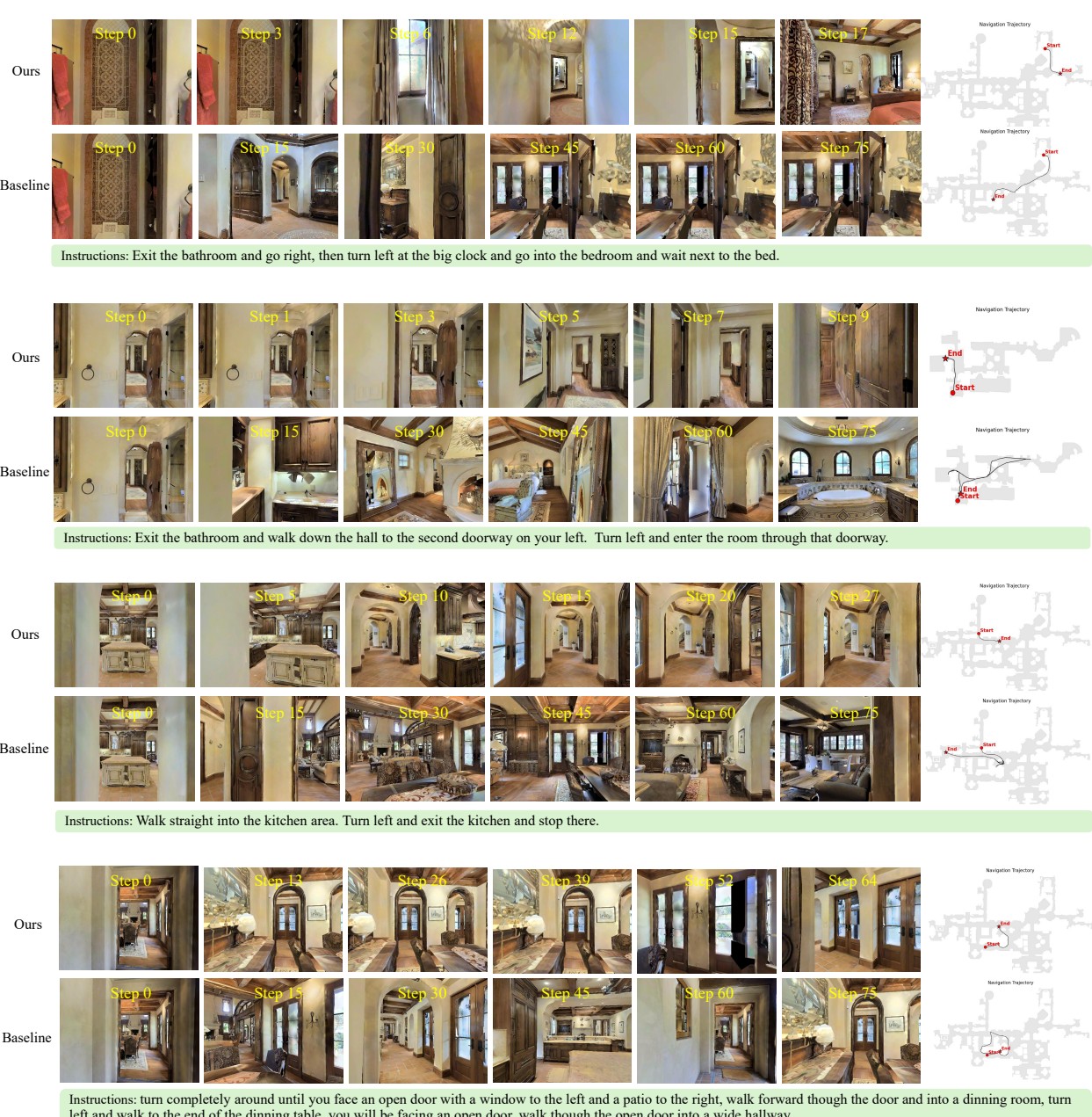

Figure 6: More visualizations for the trajectories and intermediate observations of our model against our baseline(no memory).

