# OpenReview forum: "MemoryVLN: Memory-Augmented Vision-Language Navigation"
_TMLR — Under review for TMLR_

### Review · Reviewer_d1Q8 · 2026-07-01

**Summary Of Contributions:**

The paper proposes MemoryVLN, a memory-augmented framework for continuous Vision-Language Navigation. Instead of feeding only the most recent RGB frames to a vision-language model, the method introduces three memory components: a Temporal Reasoning Memory that keeps recent frames and selects distinctive long-range frames, a Spatial Intelligence Memory that constructs a textual scene graph from detected objects and their spatial relations, and a Trajectory Memory that renders the agent’s past actions as a top-down trajectory image. These memory representations are concatenated with the current observation and language instruction, then used by an LLM-based navigation model to predict future action chunks.

The main claimed contribution is that explicitly structured memory improves long-horizon navigation, spatial reasoning, and efficiency under RGB-only observations. The paper reports strong results on R2R-CE and RxR-CE, cross-dataset generalization from R2R to RxR, and improved performance on ScanQA. The empirical results suggest that the proposed memory modules can be useful, and the decomposition into temporal, spatial, and trajectory memories is intuitive and relevant to embodied navigation.

**Strengths:**
- The paper studies an important and timely problem: how to equip VLN agents with longer-horizon memory rather than relying on a short sliding window of recent frames.
- The proposed decomposition into temporal, spatial, and trajectory memories is clear and intuitive.
- The Temporal Reasoning Memory provides a reasonable way to reduce redundancy in historical visual observations by selecting more distinctive frames.
- The Spatial Intelligence Memory attempts to make spatial structure explicit through object detection and scene-graph serialization, which is well aligned with the needs of navigation.
- The Trajectory Memory is a simple but interesting representation: rendering action history as a top-down image may be more suitable for VLMs than a long textual action log.
- The paper provides extensive empirical results on R2R-CE, RxR-CE, cross-dataset generalization, ScanQA, and several ablation studies.
- The reported gains are large, and the qualitative visualizations are helpful for understanding how the memory modules may affect navigation behavior.

**Weaknesses:**

- There is an apparent ambiguity between the claim that the method covers a temporal range much larger than 16 frames and the implementation description that short-term and long-term memory together contain 16 frames.
- The ScanQA experiment is not sufficiently explained. It is unclear how the navigation model is adapted to ScanQA, what inputs are used, and whether the comparison to 3D scene-language baselines is fair.
- Some numerical and presentation issues reduce confidence in the experimental reporting, such as the unusual Oracle Success value in Table 11 and several formatting artifacts.
- The dependence on external detectors and hand-crafted scene-graph serialization weakens the claim that the gains come purely from the proposed memory framework.
- The paper would benefit from stronger token-matched and compute-matched baselines, statistical reliability across seeds, and a more careful analysis of failure cases.

**Audience:**

Yes

**Audience Explanation:**

Yes. Vision-Language Navigation, embodied AI, memory-augmented multimodal models, and long-horizon spatial reasoning are all relevant topics for the TMLR audience. The paper’s central idea, that VLN agents should maintain structured temporal, spatial, and trajectory memories rather than rely on a short sliding window of visual observations—is interesting and timely. Researchers working on embodied navigation, multimodal foundation models, robot memory, and efficient long-context reasoning would likely find the proposed decomposition and empirical observations useful.

**Broader Impact Concerns:**

N/A.

**Claims And Evidence:**

No

**Claims Explanation:**

The submission presents promising empirical results, but I do not think the evidence is currently accurate, convincing, and clear enough to support the main claims.
First, the comparison to prior work is not sufficiently controlled. The paper claims state-of-the-art performance and emphasizes that MemoryVLN, using only RGB observations, outperforms several methods that use panoramic observations, depth, odometry, or waypoint predictors. However, the compared methods differ substantially in backbone model, training data, pretraining scale, action prediction setup, and available external modules. MemoryVLN is built on a large LLM/VLM-style model and uses additional components such as Grounding DINO for object detection and scene-graph construction. Without a controlled comparison under matched backbones, matched training data, and matched compute budgets, it is hard to attribute the gains specifically to the proposed memory design.

Second, the efficiency claim is incomplete. The paper argues that MemoryVLN is more efficient because it reduces LLM input tokens relative to feeding 16 full frames. However, this accounting appears to ignore the additional cost of running the visual encoder over historical frames, applying Grounding DINO, extracting RoI features, constructing scene graphs, rendering trajectory images, and processing the additional memory image. End-to-end latency, FLOPs, GPU memory, and wall-clock inference speed are not reported. Therefore, the claim of “high computing efficiency” is not sufficiently supported.

Overall, the evidence is suggestive, but the current paper does not yet support the claim with clear evidence.

**Requested Changes:**

Please refer to my weaknesses part.

---

### Review · Reviewer_ZEf9 · 2026-07-16

**Summary Of Contributions:**

MemoryVLN introduces a memory-augmented framework for Vision-Language Navigation (VLN), where an embodied agent must follow natural language instructions to navigate through unseen environments. The authors argue that existing VLN models rely on short histories of recent observations, limiting their ability to reason over long trajectories and understand spatial layouts. To address this, MemoryVLN incorporates three complementary memory modules: a Temporal Reasoning Memory, a Spatial Intelligence Memory and a Trajectory Memory.

Experiments on the R2R-CE and RxR-CE navigation benchmarks show that MemoryVLN achieves state-of-the-art performances. Ablation studies confirm that each memory component contributes to the performance gains, with temporal memory having the largest impact. Overall, the paper shows that structured memory significantly improves navigation efficiency, spatial understanding, and long-term decision-making in embodied agents.


**Strengths**:
- Paper clearly written
- I don't know much of this field, but the incorporated modules seems to improve the performances of the model, as demonstrated on different benchmarks, and through ablation studies.
- A lot of convincing experiments, demonstrating better results than baselines.

**Audience:**

Yes

**Audience Explanation:**

Yes, this is a topic that would interest TMLR's audience.

**Claims And Evidence:**

Yes

**Claims Explanation:**

Yes, the claims seem to be supported through the good results on the different experiments.

**Requested Changes:**

The paper seems good as it is. I don't have any suggestion of improvements. However, this might come from me not knowing anything about this area of research.

---

### Review · Reviewer_NXLV · 2026-07-19

**Summary Of Contributions:**

This paper presents MemoryVLN, a memory-augmented framework for monocular vision-language navigation. It introduces three components: a Temporal Reasoning Memory that retains recent and distinctive historical frames, a Spatial Intelligence Memory that converts detected objects and their relations into textual scene graphs, and a Trajectory Memory that visualizes past actions as a top-down path. The paper reports strong results on R2R-CE and RxR-CE, together with cross-dataset, ScanQA, ablation, and qualitative evaluations. The modular design is intuitive and the reported navigation improvements are promising. However, some claims concerning 3D understanding and computational efficiency are stronger than the evidence provided, and the paper contains quantitative inconsistencies that must be resolved.

**Audience:**

Yes

**Audience Explanation:**

Efficient long-horizon memory is an important problem in embodied navigation, and the proposed combination of temporal, spatial, and trajectory representations is likely to interest researchers working on VLN and multimodal embodied agents. If the reported results and current inconsistencies are resolved, the findings would be useful to this audience.

**Claims And Evidence:**

No

**Claims Explanation:**

The navigation results and ablations provide useful evidence that the proposed memory components may improve performance. However, three central issues prevent the evidence from being fully convincing.

First, the claim of 3D spatial understanding appears overstated: SIM operates on 2D object boxes from the current image, while the ScanQA training and evaluation protocol is insufficiently described to establish genuine 3D reasoning.

Second, computational efficiency is supported only through an approximate reduction in LLM input tokens. The additional costs of Grounding DINO, object-feature extraction, scene-graph construction, and trajectory-image encoding are not measured, so end-to-end efficiency has not been demonstrated.

Third, the paper contains direct consistency errors. In Table 11, OS is reported as 37.5 while SR is 65.4, which is impossible under the standard metric definitions; moreover, Figure 2 appears to invert several spatial relations: according to the reported bounding boxes, the flowerpot is left/above the door and the sofa bed is left/below the door, whereas the generated text states the opposite. Please clarify how these relations are computed and whether the same issue affected the SIM inputs used in the experiments.

**Requested Changes:**

All of the following are critical for acceptance:

Revise or substantiate the 3D-understanding claim. Clearly describe the ScanQA training and evaluation protocol and provide evidence that the model performs genuine 3D reasoning; otherwise, narrow the claim to single-view 2D object-relation understanding.

Evaluate end-to-end computational efficiency. Report latency, throughput, and memory usage on identical hardware while including all memory-processing components, or restrict the claim to reduced LLM input-token count.

Correct and audit the reported results. Resolve the impossible OS/SR values in Table 11 and the apparent relation errors in Figure 2, and clarify whether the latter affected the inputs used in the reported experiments.